# *In vitro* fish mucosal surfaces producing mucin as a model for studying host-pathogen interactions

**Macarena P. Quintana-Hayashi, Kristina A. Thomsson Hulthe, Sara K. Lindén**⬚*

Department of Medical Biochemistry and Cell biology, Institute of Biomedicine, Sahlgrenska Academy, University of Gothenburg, Gothenburg, Sweden

* sara.linden@gu.se

## Abstract

Current prophylactic and disease control measures in aquaculture highlight the need of alternative strategies to prevent disease and reduce antibiotic use. Mucus covered mucosal surfaces are the first barriers pathogens encounter. Mucus, which is mainly composed of highly glycosylated mucins, has the potential to contribute to disease prevention if we can strengthen this barrier. Therefore, aim of this study was to develop and characterize fish *in vitro* mucosal surface models based on commercially available cell lines that are functionally relevant for studies on mucin regulation and host-pathogen interactions. The rainbow trout (*Oncorhynchus mykiss*) gill epithelial cell line RTgill-W1 and the embryonic cell line from Chinook salmon (*Oncorhynchus tshawytscha*) CHSE-214 were grown on polycarbonate membrane inserts and chemically treated to differentiate the cells into mucus producing cells. RTGill-W1 and CHSE-214 formed an adherent layer at two weeks post-confluence, which further responded to treatment with the γ-secretase inhibitor DAPT and prolonged culture by increasing the mucin production. Mucins were metabolically labelled with *N*-azidoacetylgalactosamine 6 h post addition to the *in vitro* membranes. The level of incorporated label was relatively similar between membranes based on RTgill-W1, while larger interindividual variation was observed among the CHSE *in vitro* membranes. Furthermore, *O*-glycomics of RTgill-W1 cell lysates identified three sialylated *O*-glycans, namely Galβ1-3 (NeuAcα2–6)GalNAcol, NeuAcα-Galβ1-3GalNAcol and NeuAcα-Galβ1-3(NeuAcα2–6)GalNAcol, resembling the glycosylation present in rainbow trout gill mucin. These glycans were also present in CHSE-214. Additionally, we demonstrated binding of the fish pathogen *A. salmonicida* to RTgill-W1 and CHSE-214 cell lysates. Thus, these models have similarities to *in vivo* mucosal surfaces and can be used to investigate the effect of pathogens and modulatory components on mucin production.

## Introduction

Fish health plays an important role in aquaculture where the high fish density and continuous exposure to water with potential to contain pathogens increase the opportunities for pathogen

**Data Availability Statement:** Raw MS data on O-glycan analyses are available on glycopost (https://glycopost.glycosmos.org/) using the project ID: GPST000334 (CHSE-214) and GPST000335

(RTgillW1). MS/MS data with tentative structures are available from the UniCarb-DR repository (https://unicarb-dr.glycosmos.org/references/523).

**Funding:** This work was supported by Torvald och Britta Gahlins foundation (2019), the Swedish Research Council Formas (2018-01419), the Swedish research council (2019-01152) and the Engkvists (2015/108) and Carl Tryggers foundations (CTS 23:2971). All funds were granted to Sara Lindén. The funders had no role in study design, data collection and analysis, decision to publish, or preparation of the manuscript.

**Competing interests:** The authors have declared that no competing interests exist.

contact and entry [1]. The fish mucosal epithelia are an important line of defense against pathogens as it creates a physical barrier and contains goblet cells which secrete mucus. Mucin glycoproteins are the main components of mucus, and the mucin glycosylation profile has been described in healthy Atlantic salmon, rainbow trout and arctic charr skin, gill, and intestine [2–7]. Identification of the mucin *O*-glycome in Atlantic salmon revealed over 200 mucin glycan structures differentially distributed among gill, skin, pyloric ceca, and proximal and distal intestine [8]. Atlantic salmon skin mucins have smaller, predominantly linear glycan structures, with a less diverse glycan repertoire compared to mucin *O*-glycans from gills or intestine [8]. Fifty-four *O*-glycans have so far been described on rainbow trout mucins, consisting of up to nine monosaccharide residues [7]. Similar to Atlantic salmon, rainbow trout skin glycans were shortest on average and less diverse than glycans from gills and intestine [7]. Overall, rainbow trout and Atlantic salmon *O*-glycosylation were similar regarding differences between epithelia, core distribution, average size, and overall acidity, although specific epitopes varied between species [2–5, 7]. Mucin glycosylation and expression can change during infection and inflammation [9–12], affecting interactions with the invading pathogens, regulating pathogen growth and virulence [13–16]. In Arctic charr, intestinal inflammation results in mucin glycosylation changes that affect the interactions with the bacterial pathogen *Aeromonas salmonicida* [3]. Also, several salmonid pathogens bind to Atlantic salmon and rainbow trout mucins [17], which regulate pathogen adhesion and growth in a glycosylation dependent manner [7, 15, 18]. In mammalian contexts, glycosylation has been more extensively studied and shown to be important in cell-cell interaction, immunological responses and host-microbe symbiosis/dysbiosis [19–22].

Despite mucins being known to be of major importance for host-pathogen interactions, mucin regulation studies are hampered, as most of the mucin molecule is comprised by carbohydrate and expression of mucins thereby subjected to major post translational regulation [23]. Analyzing mucin mRNA levels as a measure of mucin production can therefore be misleading [24]. Indeed, we have frequently found a lack of correlation between these parameters [25]. Histology on its own, without metabolic labelling, can also be deceptive: the important functions of mucus are mainly carried out after being secreted, and biosynthesis and secretion rate are important parameters in this context [26]. Most histological methods lose the secreted mucus and mainly evaluate the amount of mucus inside the goblet cells. In a situation where the mucin production rate is unchanged, a decreased secretion could lead to an increased amount of mucus in the cells, conversely, an increased secretion could lead to a decreased amount of mucus in the cells. The amount of mucus within the goblet cell is thus not strongly linked to mucus production rate. Instead, metabolic labelling of mucin is a good alternative [27]. Metabolic labelling of mucins using the *N*-acetylgalactose (GalNAc) analogue *N*-azidoacetylgalactosamine (GalNAz), which incorporates into the mucin glycan core region, has been used extensively in mice, human cell lines, and in a previous study using small rainbow trout [4, 28, 29]. As a drawback, metabolic labelling is relatively expensive and therefore not suitable for studies in large fish, however, the smaller format of *in vitro* mucosal surfaces is suitable for this type of studies.

Fish and mammalian epithelial cells secrete mucins *in vivo*. However, epithelial cells grown *in vitro* under standard cell culture conditions often fail to form a tight epithelium, polarize, and secrete mucins [30]. Thus, they are deficient in the protective features typical of epithelial cells *in vivo*. We have shown that some mammalian epithelial cell lines produce mucus under certain culture conditions [30, 31]. For example, the HT29-MTX-E12 cell line cultured on semi-permeable membranes, treated with *N*-[(3,5-Difluorophenyl)acetyl]-L-alanyl-2-phenyl] glycine-1,1-dimethylethyl ester (DAPT) under semi-wet interface with mechanical stimulation for three weeks, result in an *in vitro* mucosal surface with tight epithelia and cell differentiation into a goblet cell phenotype that produces mucus [31, 32].

Fish cell lines and primary cell cultures, mainly derived from rainbow trout (*Oncorhyncus mykiss*), have been used in a variety of applications such as in toxicity testing, osmoregulatory mechanisms and environmental monitoring [33–35]. The gill epithelial cell line derived from adult rainbow trout, RTgill-W1, has been grown in cell layers responsive to tight junction modulators and osmoregulatory hormones, the latter increasing the mRNA expression of tight junction proteins claudin-10e and Cldn-30 [33]. The rainbow trout intestinal epithelial cell line RTgutGC has been used to develop a fish intestinal epithelial barrier model, where cells are cultured onto permeable membranes and tight junctions and desmosomes can be observed by electron microscopy after three weeks of culture [36]. The RTgutGC *in vitro* model has been used for the evaluation of feed ingredients on immune and barrier function, where expression of tight junction proteins and transepithelial electrical resistance can be assessed [37]. Additionally, an increase in the expression of barrier function related genes such as *claudin-3* (*cldn3)* and *interleukin*(*il)-1β*, *il-6*, *il-8* and *tumor necrosis factor-α* proinflammatory cytokines has been shown in RTgutGC cells after exposure to lipolysaccharide [37]. Advancements on the RTgutGC barrier model were made by the introduction of intestinal fibroblast cells (RTgutF) further mimicking the intestinal architecture [38]. Despite these models, there is lack of information regarding fish cell line mucin production and glycosylation.

The purpose of this study was to develop *in vitro* mucin producing mucosal surface models relevant for the study of fish-pathogen interactions in economically important salmonid species using the commercially available salmonid cell lines RTgill-W1 and CHSE-214. To achieve this, we investigated their i) ability to adhere to polycarbonate semi-permeable membranes, ii) response to agents that have been shown to induce mucin production in other contexts, iii) ability to incorporate the GalNAz metabolic label, iiii) glycan repertoire and iiiii) *A. salmonicida* binding.

## Materials and methods

### Fish cell lines

The rainbow trout (*Oncorhynchus mykiss*) epithelial cell line RTgill-W1 derived from gills was acquired from the American Type Culture Collection (ATCC, Manassas, VA, USA). The Chinook salmon (*Oncorhynchus tshawytscha*) embryonic cell line CHSE-214 was obtained from Sigma-Aldrich (Darmstadt, Germany).

### Culture of the fish cells

All fish cell lines were routinely expanded in cell culture flasks containing basic cell culture medium composed of Leibovitz's L-15 (FisherScientific, USA) supplemented with 10% (v/v) fetal bovine serum (FBS, Gibco, FisherScientific, USA) and 1% (v/v) penicillin-streptomycin (P/S, Lonza, Switzerland) and incubated at 21 ± 5°C inside a cell incubator with atmospheric air. Subconfluent cultures (70–80%) were rinsed with Dulbecco's phosphate buffered saline (w/o $Ca^{++}$ and $Mg^{++}$) and depending on the size of the cell culture flask, 1.5–3 mL of Trypsin-EDTA (Gibco, FisherScientific, USA) was added and incubated for 5–10 min at 21 ± 5°C to detach cells from the flask. The cells were harvested in basic cell culture medium and centrifuged at 1,000 RCF for 5 min. The cell pellet was resuspended in cell culture medium and the number of viable cells was determined using the Trypan Blue exclusion technique. The cells were diluted accordingly to obtain $7.5 \times 10^4$ cells per 200 μL.

## Chemical stimulation of the fish cells

Two hundred μL with $7.5 \times 10^4$ cells were plated on the apical side of the Snapwell cell culture inserts (12 mm diameter 0.4 μm pore polycarbonate membrane supported by a detachable ring, Corning Costar, New York, USA) and 4 mL of basic cell culture medium was added in the basolateral compartment. The cells were placed inside a cell incubator at $21 \pm 5°C$ with atmospheric air, and the basolateral medium was changed until the cells were confluent.

Three types of differentiation media were used to attempt to differentiate the cell lines into producing more mucus. Together with the basic cell culture medium control, we thus had four different set ups: 1) basic cell culture medium control, 2) addition of 10 μM of the γ-secretase inhibitor DAPT (Sigma-Aldrich, Darmstadt, Germany) to the basic cell culture medium, 3) addition of 2 μM of the Wnt pathway inhibitor IWP-2 (Stemcell Technologies, Cambridge, UK) to basic cell culture medium and 4) addition of 10 μM DAPT and 2 μM of IWP-2 to the basic cell culture medium. The DAPT and IWP-2 were added to the basolateral compartment for six days, changing the differentiation medium (basic medium containing DAPT and or IWP-2) daily. DAPT is a δ-secretase inhibitor that blocks the Notch signaling pathway while IWP-2 inhibits the Wnt signaling pathway, driving cell differentiation towards a secretory line-age, including goblet cell differentiation. After 6 days the differentiation medium was replaced with basic cell culture medium, changing the medium every 2–3 days in the beginning and daily towards the end. The position of the plates inside the incubator was shuffled after every change of medium. The cells were harvested after two- and four-weeks post-confluence. Six *in vitro* membranes were analyzed per treatment and time point, and the experiments were repeated twice.

## GalNAz labeling of mucins

DAPT treated fish cells grown on polycarbonate membranes for four-weeks post confluence were treated with GalNAz for mucin labelling. The Click-IT GalNAz metabolic glycoprotein labelling reagent (ThermoFisher Scientific, Waltham, MA, USA) was dissolved in DMSO and diluted in basic cell culture medium to a final concentration of 240 μM. The medium in the basolateral compartment was replaced with 4 mL of basic cell culture medium containing Gal-NAz 6 hours (h) prior to harvest of the membranes.

## Preservation of the *in vitro* mucosal surfaces

The membranes were 'sandwiched' and fixed in methanolic Carnoy's fixative. For this purpose, one of the membranes was cut out from its insert using a surgical scalpel blade and with the help of a small forceps carefully placed on top of another membrane still in its insert, making sure that both apical surfaces of the membranes were facing each other. The 'sandwiched' membranes were placed in Carnoy's fixative (60% (v/v) dried methanol (max. 0.003% $H_2O$), 30% (v/v) chloroform and 10% (v/v) acetic acid) for 24–48 h. Fixation was followed by immersion in dry methanol for 30 min. and paraffin infiltration in a tissue processor starting with two changes of xylene (10 min each), followed by three changes of paraffin wax (15 min each). The membranes were embedded in paraffin blocks and sectioned with a microtome for subsequent staining.

## PAS/alcian blue staining

Paraffin embedded sections were deparaffinized and rinsed in water for 10 min, immersed in 3% acetic acid for 2 min, and stained in 1% Alcian blue 8GX (Merck, Darmstadt, Germany) diluted in 3% acetic acid (Merck, Darmstadt, Germany) pH 2.5 for 2.5 h. The sections were

then dipped in 3% acetic acid and rinsed in water for 10 min. Oxidation was performed in 1% periodic acid for 10 min, washed in water for 5 min, immersed in Schiff's reagent (Merck, Darmstadt, Germany) for 15 min, rinsed in water for 5 min, and rinsed in 0.5% sodium meta-bisulphite (Merck, Darmstadt, Germany) three times (1 min each), followed by a final wash in water for 5 min. The sections were then dehydrated and mounted.

### Fluorescent detection of GalNAz

Paraffin embedded membrane sections of DAPT treated fish cells grown on polycarbonate membranes for four-weeks post confluence and treated with GalNAz for mucin labelling (and non-labelled sections, negative controls), were deparaffinized, hydrated in 30% ethanol and washed in PBS. Twenty microliters of the TAMRA (tetramethylrhodamine) glycoprotein detection kit (ThermoFisher Scientific, Waltham, MA, USA) were added on top of each section and incubated in the dark for 2 h in a humidified chamber at room temperature (approximately 20°C). The membrane sections were washed in PBS and mounted with ProLong® antifade containing DAPI (ThermoFisher Scientific, Waltham, MA, USA). Fluorescence intensity measurement data was obtained for a total of six membranes per cell line with a NIS-Elements D v. 4.0 software.

### Cell lysate extraction

RTgill-W1 and CHSE-214 cells were cultured in Leibovit's L-15 medium supplemented with 10% (v/v) FBS and 1% (v/v) penicillin-streptomycin (P/S). Cells were incubated at $21 \pm 5$°C inside a cell incubator with atmospheric air for two weeks post-confluence, changing the cell medium every 2–3 days. The cells were washed with PBS and scraped from the cell culture flask in 500 µL of extraction buffer (6 M guanidine hydrochloride (GuHCl), 5 mm EDTA, 10 mm sodium phosphate buffer (pH 6.5) containing 0.1 mM of PMSF (PanReac AppliChem, Germany). The cells were homogenized by four strokes with a loose pestle using a Dounce homogenizer. The homogenized solution was transferred to a new tube together with another two sample volumes of extraction buffer used to rinse the homogenizer. Samples were put on a rocking board for 20 h at 4°C, followed by centrifugation at $3900 \times g$ for 80 min to remove cell debris. Supernatants were collected.

### *O*-glycan release from cell lysates on dot blots

Aliquots of cell lysate (above) were dot blotted to PVDF membrane (Immobilon P, Millipore) and visualized using alcian blue stain. Glycans were released using a modified protocol published elsewhere [39]. PVDF membrane spots were excised and placed in test tubes followed by 5 x 15 min destain/washes in MeOH. The *O*-glycans were released from the protein with 40 µL beta elimination solution (1 M NaBH4 in 0.1 M NaOH) at 50°C in a water bath. Samples were neutralized with 1–2 µL acetic acid, followed by desalting using cation exchange media (AG50WX8 (Biorad) in C18 ziptips (Millipore), two ziptips/sample, and dried with vacuum centrifugation. Borate residuals were eliminated by repeated additions of MeOH (5 x 50 µL) and dried in between.

### SDS-PAGE and *O*-glycan release from gel bands

Cell lysate aliquots were dialyzed using 14 kD cut off dialysis tubes (D9277, Sigma-Aldrich), and lyophilized. Cell lysates and control media were reduced in 25 mM iodacetamide (ThermoScientific, USA) for 30 min at 90°C, followed by alkylation with 63 mM iodacetamide in LDS sample buffer (ThermoFisher). Samples were analyzed on Tris-acetate gels (3–8%,

NuPage, ThermoFisher) run at 4˚C. Molecular marker was Pageruler Plus prestained Protein ladder (Thermo Scientific). Gels were developed in Alcian blue, and bands cut out for *O*-glycan release using a modified protocol [40]. Briefly, gel pieces were washed by repeated extractions with ethyl acetate, and glycans released using 1 M NaBH4 in 0.1 M NaOH.

## Liquid chromatography mass spectrometry (LC/MS) of *O*-glycans

Reduced *O*-glycans were resuspended in 15 μL of water and injected (2 μL) onto a liquid chromatography-electrospray ionization tandem mass spectrometry (LC-ESI/MS) using a UltiMate 3000 pump and autosampler connected to a low resolution linear ion trap (Thermo Scientific) run in the negative ion mode. The oligosaccharides were separated on a HPLC column (10 cm × 250 μm id) packed in-house with 5 μm porous graphite particles (PGC, Hypercarb, Thermo-Hypersil, Runcorn, UK) and a flow rate of 5 μL/min. The oligosaccharides were eluted with the following gradient: 0–46 min 0–45% B, wash 46–54 min 100% B, then equilibration between 54–78 min with 0% B. Buffer A was 10 mM ammonium bicarbonate (ABC), and buffer B was 10 mM ABC in 80% acetonitrile.

A mass range of *m/z* 380–2000 was used and MS/MS was made at a normalized collisional energy of 35%. Compressed air was used as nebulizer gas. The stainless-steel needle was kept at -2.5 kV. The isolation width was set to *m/z* 3.0 with an activation time of 30 ms and a minimal signal threshold of 300 counts for MS/MS. Glycans were detected as [M-H]⁻ ions and identified by comparing MS$^2$ spectra to published reference spectra [5, 7]. The glycan MS signals were quantified using Xcalibur software (version 2.0.7, Thermo Scientific, Waltham, MA, USA). Raw MS data on *O*-glycan analyses are available on glycopost (https://glycopost.glycosmos.org/) using the project ID: GPST000334 (CHSE-214) and GPST000335 (RTgillW1). MS/MS data with tentative structures are available https://unicarb-dr.glycosmos.org/references/523.

## Bacterial culture

*Aeromonas salmonicida* subsp. *salmonicida* strain VI-88/ 09/03175 (culture collection, Central Veterinary Laboratory, Oslo, Norway) was cultured on tryptone soy agar (TSA) plates and incubated at 20˚C.

## *A. salmonicida* binding to RTgillW1 and CHSE-214 cell lysates

Cell lysates were dialyzed using 14000 kD cut off dialysis tubes (D9277, Sigma-Aldrich) and used at a concentration of approximately 0.3 μg/mL. Twenty-five μL were loaded in duplicate onto a PVDF-FL membrane (MilliporeTM) using a Slot Blot apparatus (Minifold–II, Schleicher & Schuell Bioscience GmbH). Twenty-five μL of dialyzed L-15 cell medium and PBS were loaded as controls. The membranes were then air-dried for 1 h, pre-wetted briefly in 100% methanol, rinsed with distilled water and incubated in PBS for 10 min. The membranes were blocked with Odyssey blocking buffer (LI-COR BiosciencesTM) for 1 h. *A. salmonicida* was biotinylated as previously described [41], and membranes were incubated with biotinylated bacteria at an optical density (600 nm) of 0.1 diluted 1:20 in Odyssey blocking buffer containing 0.1% Tween 20 for 2 h at room temperature with gentle shaking. Membranes were washed three times for 5 min with PBS-T (0.2% Tween20) and incubated with streptavidin labelled IR dye 800 (LI-COR, BiosciencesTM) diluted 1:10,000 in Odyssey blocking buffer containing 0.1% Tween20 for 30 min. Membranes were washed three times for 5 min with PBS-T (0.2% Tween20) followed by a final rinse with PBS. The blots were visualized with an Odyssey CLx infrared imaging system (LI-COR, BiosciencesTM).

## Statistics

The Kolmogorov-Smirnov test was used to test data sets for normality. Students T-test was used to compare datasets. The two sample F-test for equality of variances was used to test if the variances differed between data sets. The standard deviation, SD, of datasets were used to calculate the sample variances ($SD^2$). A p-value below 0.05 was considered statistically significant.

## Results

### RTgill-W1 and CHSE-214 cell lines form adherent layers and produce mucins

We started by investigating morphology, adherence, and mucin staining in the salmonid cell lines RTgill-W1 and CHSE-214 under standard conditions, three different goblet cell differentiating media (DAPT and IWP-2, separately and combined) and prolonged culture. At two and four-weeks after the cells were confluent, the membranes with cells grown on them (here after referred to as *in vitro* membranes) were harvested for histology and mucin carbohydrates stained with PAS/Alcian blue. Cells from all treatments and time points had at least some Alcian blue positive material but no visible PAS stain, suggesting that the cell lines mainly produce acidic mucins. At two weeks post-confluence RTgill-W1 and CHSE-214 cells had formed a thin and continuous adherent layer on the apical surface of the polycarbonate membranes, with RTgill-W1 cells having a flat appearance compared to the more rounded CHSE-214 cells (Fig 1A and 1F). However, no effects of the treatments were observed regarding staining of mucin carbohydrates at two weeks post-confluence, in either RTgill-W1 or CHSE-214 cells, i.e. *in vitro* membranes from all treatments appeared similar to the non-treated *in vitro* membrane shown in Fig 1 (panels A and F). In comparison, at four weeks post-confluence, differences between treatments were observed. Alcian blue staining increased in both RTgill-W1 and CHSE-214 cells treated with DAPT (Fig 1C and 1H) compared to the non-treated cells (Fig 1B and 1G), suggesting that DAPT increased production of acidic mucins. Macroscopically the apical medium of DAPT treated cells presented a slimy mucous-like consistency. In

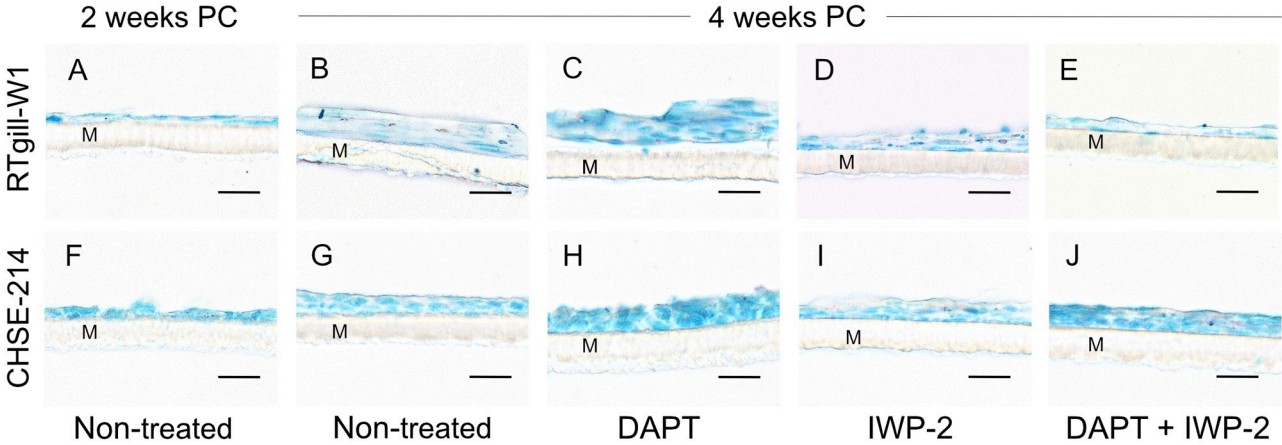

**Fig 1. RTgill-W1 and CHSE-214 adherent layers.** Representative images of PAS/Alcian blue stained RTgill-W1 (A-E) and CHSE-214 (F-J) cells grown on polycarbonate membranes (M) at two (A and F) and four (B-E, G-J) weeks post-confluence (PC). (A and F) Non-treated cells harvested at 2 weeks post-confluence. (B and G) Non-treated cells harvested at four weeks post-confluence. (C and H) DAPT treated cells harvested at four weeks post-confluence. (D and I) IWP-2 treated cells harvested at four weeks post-confluence. (E and J) Cells treated with a combination of DAPT and IWP-2 harvested at four weeks post-confluence. Images were captured with an Eclipse 90i microscope (Nikon) and are representative of six replicates per treatment. Scale bar = 30 μm.

contrast, treatment with IWP-2 did not have a major impact on PAS/Alcian blue staining, and the cell-layers were similar in appearance to the non-treated cell-layers at four weeks post confluence (Fig 1D and 1I). Treatment with IWP-2 in combination with DAPT negated the goblet cell inducing effects of DAPT on the RTgill-W1 cell-layers, as it, in appearance, resembled the thin adherent layer observed at the two-week post confluence time point (Fig 1E). On the other hand, in CHSE-214 cells, the combined treatment of DAPT and IWP-2 enhanced mucus staining compared to IWP-2 treatment alone and the non-treated control (Fig 1G, 1I and 1J). This was potentially attributed to the enhancing effect of DAPT (Fig 1H).

## The metabolic label GalNAz is incorporated into both RTgill-W1 and CHSE-214 based *in vitro* membranes

Metabolic labelling of mucins with GalNAz can be used as a measure of mucin production. At six h post addition of GalNAz to the basolateral compartment of the tissue culture wells, incorporated GalNAz was clearly visible in the majority of both RTgill-W1 and CHSE-214 cells (Fig 2). There was no statistically significant difference in the level of GalNAz incorporation between the cell lines (p = 0.11). In the RTgill-W1 *in vitro* membranes, the level of incorporated GalNAz was relatively similar between membranes (Standard deviation, SD: 9.88), while a tendency to larger variation between membranes was observed among the CHSE *in vitro* membranes (SD: 20.98, two sample F-test: p = 0.12, Fig 2C).

## *O*-glycan analyses from CHSE-214 and RTgill-W1

To characterize CHSE-214 and RTgill-W1 glycosylation, *O*-glycans in cell lysates were analyzed using LC/MS. Galβ1-3(NeuAcα2–6)GalNAcol, NeuAcα2-3Galβ1-3GalNAcol, NeuAc-Gal-(NeuAc-)GalNAcol were abundant in CHSE-214 lysates, which also contained lower levels of SO3-Gal-(NeuAc-)GalNAcol, SO3-Gal-GalNAcol and Gal1HexNAc1GalNAcol (Table 1). NeuAcα2-3Galβ1-3GalNAcol and NeuAc-Gal-(NeuAc-)GalNAcol were the most

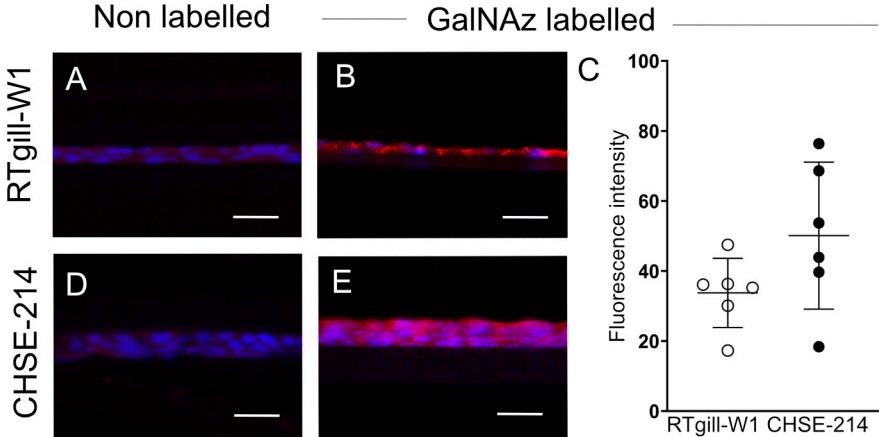

**Fig 2. Incorporation of GalNAz label in RTgill-W1 and CHSE-214 cell lines grown on polycarbonate membranes at four weeks post-confluence.** Representative images of metabolically labelled mucins (red), used as a measure of overall mucin production, in RTgill-W1 (B) and CHSE-214 cells (E), six h post addition of GalNAz, compared to cells without GalNAz addition (A and D, negative controls). N = 6, scale bar = 30 μm. Images were captured with an Eclipse 90i microscope (Nikon). (C) Fluorescence intensity measurement data was obtained for a total of six membranes per cell line with a NIS-Elements D v. 4.0 software. The quantified fluorescence intensity data sets for both cell lines passed the Kolmogorov-Smirnov test for normality. The sample variances (SD$^2$) were 97.55 and 440.11 for RTgill-W1 and CHSE-214, respectively. The fluorescence intensity did not differ between RTgill-W1 and CHSE-214 (T-test, p = 0.11). Cells were counterstained with DAPI (blue).

**Table 1. *O*-glycans from RTgill-W1 and CHSE-214 cell lines detected by LC/MS.**

| Name | Precursor ion (m/z) | Putative glycan | Cell media (dot blot) | CHSE-214 | | | RTGillW1 | | |
|---|---|---|---|---|---|---|---|---|---|
| | | | | Cell lysate (dot blot) | C2' gel band | | Cell lysate (dot blot) | R1' gel band | R2' gel band |
| 464a | 464 | Gal-(SO3-)GalNAcol | - | - | 72% | | - | - | 100% |
| 464b | 464 | SO3-Gal-GalNAcol | - | 9% | 28% | | 4% | - | - |
| 587 | 587 | Gal1HexNAc1GalNAcol | - | 4% | - | | - | - | - |
| 675a | 675 | Galb1-3(NeuAca2-6)GalNAcol | 2% | 20% | - | | 6% | 33% | - |
| 675b | 675 | NeuAca2-3Galb1-3GalNAcol | 80% | 33% | - | | 45% | 25% | - |
| 755 | 755 | SO3-Gal-(NeuAc-)GalNAcol | - | 9% | - | | 12% | - | - |
| 966 | 966 | NeuAc-Gal-(NeuAc-)GalNAcol | 18% | 25% | - | | 33% | 42% | - |

Glycans were detected in the negative ion mode as [M-H]- precursor ions. MS2 spectra were compared to published reference spectra. The glycans are reported as relative abundance, based upon their MS signal.

abundant glycans from the RTgill-W1 lysates, which also contained SO3-Gal-GalNAcol, Galβ1-3(NeuAcα2–6)GalNAcol and SO3-Gal-(NeuAc-)GalNAcol (Table 1). Although the cells were washed before lysis, the media the cells were grown in was also analyzed to ensure the glycans described were from the cells and not contaminants from the media. The media contained NeuAca2-3Galβ1-3GalNAcol, NeuAc-Gal-(NeuAc-)GalNAcol and Gal1HexNAc1-GalNAcol, albeit at different proportions than the cell lysates (Table 1). To verify that the glycans originated from the cell lines (i.e. not a contaminant from the media), reduced and alkylated proteins from cell lysates were separated with SDS-PAGE for large proteins, and the gels developed with Alcian Blue, which stains for acidic glycans (Fig 3). Both cell lysates stained extensively, however there was not much stain above 250 kD, where mucins and mucin-like proteins from mammalian tissue normally are found. SDS-PAGE revealed that potential contaminating glycoproteins that could originate from cell media migrated mainly below 70 kD (Fig 3). Two gel bands per cell line were selected for further *O*-glycomics studies, collected just below the bottom of the loading well (CHSE-214: 'C1'; RTGill-W1: 'R1'), and at approximately 250 kD ('C2' and 'R2', respectively) for both cell lines (Fig 3). Three *O*-glycans were detected in the 'R1' band from RTGill-W1 (Galβ1-3(NeuAcα2–6)GalNAcol, NeuAcα-Galβ1-3GalNAcol and NeuAcα-Galβ1-3(NeuAcα2–6)GalNAcol, Table 1). *O*-glycomics of the RTGill-W1 gel band 'R2' at 250 kD revealed one sulfated *O*-glycan (Gal-(SO3-)GalNAcol, Table 1). For the CHSE-214 cell line, no glycans were found in the 'C1' band, however two sulfated disaccharides were detected in the protein band at 250 kD ('C2', Gal-(SO3-)GalNAcol and SO3-Gal-GalNAcol, Table 1). The presence of these glycans in other areas of the gel than the main bands produced by the media, together with the difference in proportion of glycans present in the media and cell lines, suggest that these glycans originate from the cells and not from the media.

## RTgill-W1 and CHSE-214 cell lysates bind *A. salmonicida*

We investigated the binding ability of *A. salmonicida* to RTgill-W1 and CHSE-214 cell lysates slot-blotted on to PVDF-FL membranes. *A. salmonicida* bound to lysates from both RTgill-W1 and CHSE-214 cells cultured for two weeks post-confluence. No binding signal was obtained to slots with un-used cell culture media (Leibovitz's L-15 supplemented with 10% FBS and 1% penicillin-streptomycin) or PBS (Fig 4).

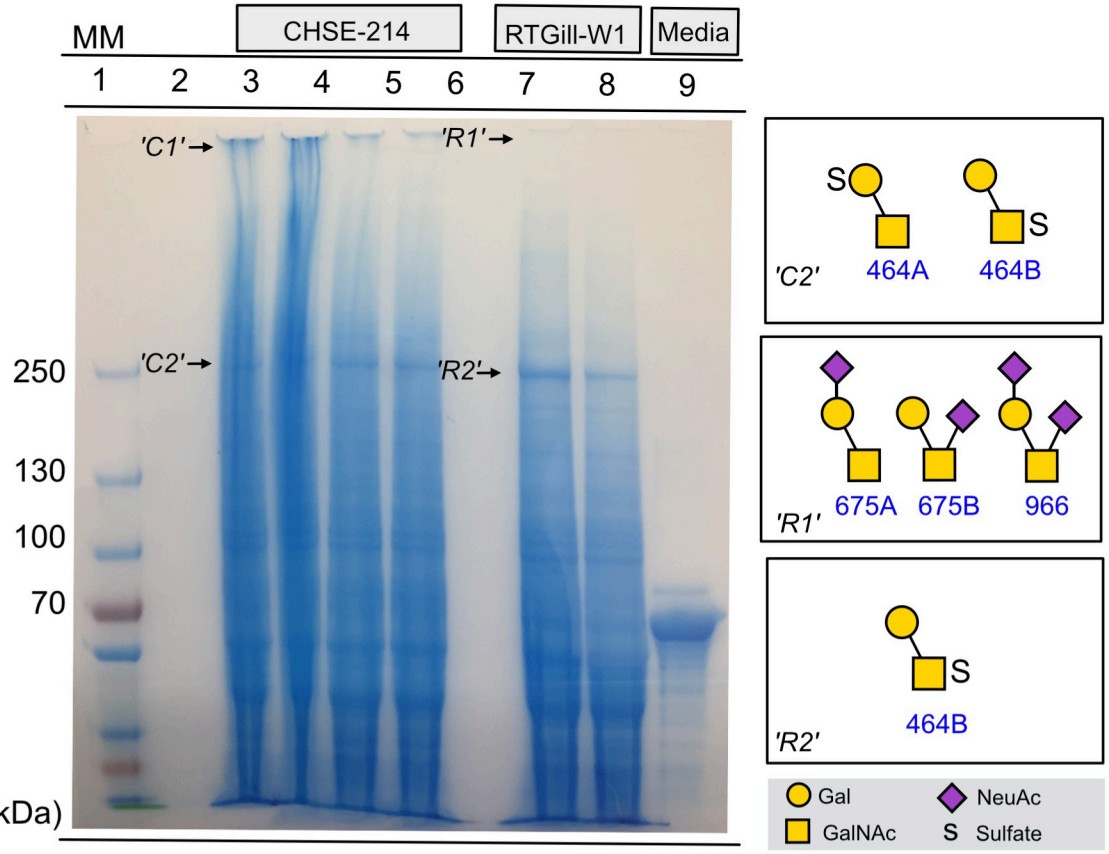

**Fig 3. *O*-glycans on high molecular weight glycoproteins from cell lysates of cell lines CHSE214 and RTGill-W1.** Cell lysates were analysed on 3–8% tris acetate gels, and developed with Alcian Blue, which stains acidic glycoproteins (The uncropped image is available as S1 Fig in S1 Raw image). *O*-glycans were released from the proteins from selected bands 'C1', 'C2' (CHSE-214),'R1' and 'R2' (RTGillW1) as shown. The *O*-glycans were analysed with LC-MS, and their relative abundance was estimated from the MS signal and summarized in Table 1.

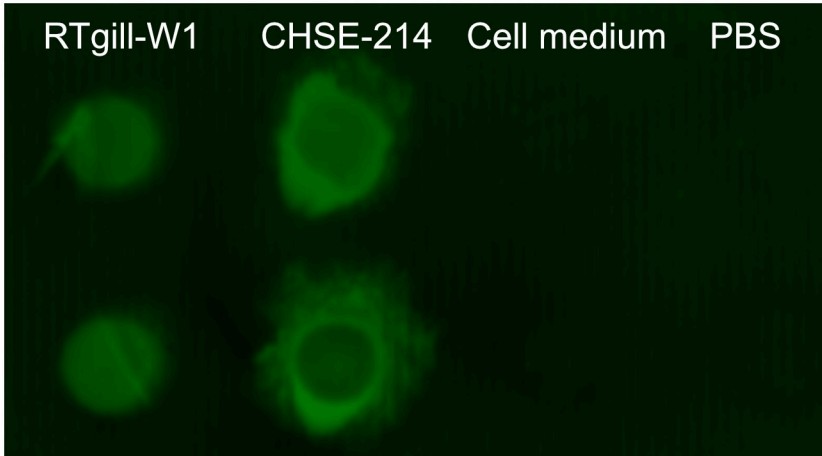

**Fig 4. *A*. *salmonicida* binding to RTgill-W1 and CHSE-214 dialyzed cell lysates on PVDF-FL membrane.** Binding is shown as green fluorescence in duplicates. Dialyzed un-used cell media (Leibovitz's L-15 supplemented with 10% FBS and 1% penicillin-streptomycin) and PBS were loaded as controls.

## Discussion

We here found that RTgill-W1 and CHSE-214 form adherent *in vitro* mucosal surfaces and that their mucin production can be enhanced by prolonged culture and DAPT but not IWP-2. Furthermore, we identified that GalNAz is incorporated in a similar pattern as the Alcian blue staining in both RTgill-W1 and CHSE-214. We characterized six *O*-glycans from RTgill-W1, which were all acidic, and seven *O*-glycans from CHSE-214, of which six were acidic. Six of the *O*-glycans were common to both cell lines. Lysates from RTgill-W1 and CHSE-214 both contained components that *A. salmonicida* bound to.

Considering that animal experiments are suboptimal because of ethical considerations, it is important to develop methods that can replace *in vivo* experiments. An alternative could be cell-based *in vitro* models that mimic teleost mucosal surfaces for experimental use. Standard culture conditions where cells are cultured unstimulated on a plastic surface, often result in non-differentiated and non-polarized cells that lack important components of the glycocalyx, and do not produce mucins [31]. In contrast, the natural situation in a body involves cell differentiation and activation of differentiating pathways. We have previously developed a mucus secreting human colonic *in vitro* mucosal model which has been used to investigate a wide array of host-pathogen interactions, including regulation of mucus secretion and mitochondrial function during infections [28, 29, 42, 43]. In this model, human colonic HT29-MTX-E12 cells are differentiated by mechanical stimulation and inhibition of gamma secretase by DAPT [32]. Mechanical stimulation of the colonic cells is provided by a gentle rocking motion that continuously wets the apical surface of the cells under semi-wet interface. Although fish are submerged in water most of the time we considered trying this method to 'trick' the cells to increase mucus production, however, the effects of a semi-wet interface with mechanical stimulation on mucin production could not be explored in the fish *in vitro* model here proposed, as a rocking platform increased the incubator temperature by 12°C, resulting in a suboptimal environment for the fish cells. Instead, the fish cells were cultured submerged in medium both on the apical and basolateral sides. Similar to the mammalian *in vitro* mucosal surfaces [28], the salmonid cell lines RTgill-W1 and CHSE-214 formed an adherent continuous layer from two weeks post-confluence, mimicking *in vivo* mucosal surfaces. Treatment of human intestinal cells with DAPT inhibits the Notch signalling pathway and induces a mucus producing goblet cell phenotype [32, 44]. Despite a lack of information about pathways regulating differentiation of mucin producing cells in rainbow trout and chinook salmon, the Notch signaling pathway appears to be highly conserved. Thus, the adherent layer formed by both salmonid cell lines further responded to DAPT treatment, enhancing both the level of Alcian blue positive material in the *in vitro* mucosal surface and giving the media on the apical side a slimy mucus like consistency.

To study mucin production, mucins can be metabolically labelled both in *in vitro* cell culture systems and *in vivo* using GalNAz [4, 28, 45]. In mammalian goblet cells *in vitro* as well as *in vivo*, the localization of the GalNAz labelled mucins can be tracked in the different cell compartments from perinuclear to apical cell surface [28, 44]. Incorporation of GalNAz into newly synthesized mucins in the perinuclear compartment of the cell can be visualized one hour after injection in the mouse stomach [45], and in the distal colon the time from incorporation of GalNAz to release of the labelled mucin into the lumen is 6–8 h [46]. This also applies to rainbow trout *in vivo*, where mucins have been metabolically labelled in the stomach and intestine, and GalNAz incorporation can be detected at 8 h post intraperitoneal injection [4]. Here, we demonstrated GalNAz incorporation into salmonid mucins in the *in vitro* membranes. Although the relatively flat morphology of the RTgill-W1 and CHSE-214 cells did not allow for analysis in different compartments within the cell, the salmonid *in vitro* membrane models

were reproducible and overall mucin production could be assessed. In the RTgill-W1 *in vitro* membranes, the level of incorporated GalNAz was relatively similar between membranes, while a tendency to larger interindividual variation was observed among the CHSE-214 *in vitro* membranes. A lower number of replicates are thus likely to be needed when using the RTgill-W1 *in vitro* membranes compared to CHSE-214 for studies comparing effects of different conditions/treatment on mucin production.

It has been described that gill mucin proteins from rainbow trout share similar characteristics with mammalian mucins, such as large size, high content of threonine and serine, and high glycosylation [47]. Furthermore, analyses of rainbow trout gill mucins with SDS-PAGE showed that these barely entered the 4% polyacrylamide stacking gel [47]. We performed *O*-glycomics on the same gel region of RTgill-W1 derived proteins separated on 3–8% tris-acetate gels, and detected three sialylated *O*-glycans: Galβ1-3(NeuAcα2–6)GalNAcol, NeuAcα-Galβ1-3GalNAcol and NeuAcα-Galβ1-3(NeuAcα2–6)GalNAcol. These glycans are abundantly present on gill mucin purified from rainbow trout and constitute 34% (average of three analyses) of the glycan pool [7], providing evidence that the RTGill-W1 cells contain mucins that may resemble gill mucins from rainbow trout. Although the majority of the rainbow trout mucins are sialylated in line with the results from the RTGill-W1 mucin glycans, approximately 30% of the rainbow trout gill glycans were fucosylated [7], a feature we did not detect among the RTGill-W1 mucin glycans. If fucosylation is an important feature for planned experiments, it is possible to transfect cell lines with glycosyltransferases, including fucosyltransferases, to obtain a required glycoprofile [48, 49].

The embryonic CHSE-214 cell line displayed an Alcian Blue stained band which barely entered the tris-acetate gel. However, no glycans were detected. That no glycans were detected from this band even after two attempts was surprising as the band appeared stronger than for RTGill-W1, and it is possible that the lack of glycan signal could be due to an unresolved technical issue. *O*-glycomics of Alcian Blue stained bands from both cell lines at 250 kD, revealed the same single sulfated disaccharide interpreted as Gal-(SO3-)GalNAcol. This glycan has not been detected before on fish mucins from rainbow trout, Atlantic salmon or Arctic charr [2–5, 7], however it has been found on human mucins [50]. The *O*-glycome of Chinook salmon has currently not been characterized.

*A. salmonicida* binds to sialic acid (*N*-acetylneuramic acid, NeuAc), especially when linked via a 2,6 linkage [15, 51], which was present in both cell lines. The high abundance of sialylated glycans is likely the reason that *A. salmonicida* bound to both cell lines. That both cell lines carry epitopes *A. salmonicida* can bind to supports that they can be used to study host-pathogen interactions. For example, it could be interesting to investigate the role of specific glycans in host pathogen interaction by gene editing the cells for specific glycosyltransferases and compare the impact of presence/absence of specific glycans on pathogen binding and downstream host responses.

Mucin production has been studied in relation to infection and stimuli. Acute and chronic bacterial infections in mice inhibit mucin production impairing clearance of pathogens from the mucosal surface [25, 45]. On the other hand, in a spontaneously clearing bacterial infection mucus production increases and bacteria loose contact with the epithelium [25]. Stimuli such as bacterial LPS is known to upregulate mucin expression and stimulate mucin secretion in goblet cells *in vitro* and *in vivo* [4, 52, 53]. In zebrafish larvae, LPS treatment stimulates secretion of the fluorescent mucin-reporter *muc5.1*:S-RFP [53]; and in rainbow trout a 24 h LPS stimulation via anal intubation enhances mucin production and transport in the intestinal epithelium [4]. The use of mucin producing cell lines to study mucin regulation would allow the researcher to investigate effects of a large range of compounds on mucin production to increase the limited knowledge on mucin regulation in teleost epithelia.

In conclusion, well characterized *in vitro* mucosal surface models present an alternative method to study host-pathogen interactions in fish, decreasing animal use in research. As fish pathogens often infect via the mucosal surfaces, a salmonid *in vitro* membrane model consisting of epithelial surfaces that secrete mucus allows for the investigation of mucus production, secretion mechanisms, and response to immune components and relevant pathogens in aquaculture. RTgill-W1 and CHSE-214 form adherent *in vitro* mucosal surfaces, their mucin production can be enhanced by prolonged culture and DAPT and they incorporate GalNAz in a similar pattern as the Alcian blue staining. Similar to the glycans produced by rainbow trout and Atlantic salmon, the majority of the glycans from both cell lines contained NeuAc, which likely is the reason that *A. salmonicida* bind to them.

## Supporting information

**S1 Raw image. S1 Fig. The uncropped image of the gel from Fig 3.**
(PDF)

**S1 Table. Fluorescence intensity dataset for Fig 2.**
(DOCX)

## Author Contributions

**Conceptualization:** Sara K. Lindén.

**Funding acquisition:** Sara K. Lindén.

**Investigation:** Macarena P. Quintana-Hayashi, Kristina A. Thomsson Hulthe.

**Project administration:** Sara K. Lindén.

**Supervision:** Sara K. Lindén.

**Visualization:** Macarena P. Quintana-Hayashi, Kristina A. Thomsson Hulthe.

**Writing – original draft:** Macarena P. Quintana-Hayashi.

**Writing – review & editing:** Sara K. Lindén.

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
