## [Decision Letter · Decision Letter 0]

4 Feb 2024

PONE-D-23-19772Mucin Producing In vitro Fish Mucosal Surfaces as a Model for Studying Host-Pathogen InteractionsPLOS ONE

Dear Dr. Lindén,

Thank you for submitting your manuscript to PLOS ONE. After careful consideration, we feel that it has merit but does not fully meet PLOS ONE’s publication criteria as it currently stands. Therefore, we invite you to submit a revised version of the manuscript that addresses the points raised during the review process.

We look forward to receiving your revised manuscript.

Kind regards,

Maria del Mar Ortega-Villaizan

Academic Editor

PLOS ONE

Journal Requirements:

3. We note that your Data Availability Statement is currently as follows: "All relevant data are within the manuscript and its Supporting Information files. Raw MS data on O-glycan analyses are available on glycopost ( https://glycopost.glycosmos.org/) using the project ID: GPST000334 (CHSE-214) and GPST000335 (RTgillW1). MSMS data with tentative structures are available https://unicarb-dr.glycosmos.org/references/523."

Additional Editor Comments:

The manuscript presented by Linden et al is of interest, however, major revisions should be performed before being aceptable for publication in PLOS One. Please follow the comments indicated by the reviewers.

Reviewers' comments:

Reviewer's Responses to Questions

**Comments to the Author**

1. Is the manuscript technically sound, and do the data support the conclusions?

Reviewer #1: Yes

Reviewer #2: No

Reviewer #3: Yes

2. Has the statistical analysis been performed appropriately and rigorously? 

Reviewer #1: N/A

Reviewer #2: No

Reviewer #3: No

3. Have the authors made all data underlying the findings in their manuscript fully available?

Reviewer #1: Yes

Reviewer #2: No

Reviewer #3: Yes

4. Is the manuscript presented in an intelligible fashion and written in standard English?

Reviewer #1: Yes

Reviewer #2: Yes

Reviewer #3: Yes

5. Review Comments to the Author

Reviewer #1: This manuscript describes an interesting novel in vitro model based on modified fish cell lines for the study of mucin responses in mucosal epithelia. Its reading was inspiring and it will help to develop alternative methods to the in vivo ones.

There are some suggestions authors could follow in order to detail results and deepen discussion, which in my opinion could increase the study’s soundness.

-I wonder what the cell morphology in the in vitro membranes was, and if it changed after DAPT treatment. Could authors observe goblet cell morphology or cell polarization? Including a description on cell morphology, differentiation and polarization would be interesting. In fact, authors have included in the discussion section the issue of non-differentiation and non-polarization in unstimulated cultures on plastic surfaces, leaving this issue unanswered in the current model.

-Authors describe interindividual variation of GalNAz incorporation in terms of “relatively similar” or “larger variation” in each cell line. Statistical significance should be indicated in terms of an analysis of variance.

-L430 and following: Authors could include a comment on the need of semi-wet interfaces for fish in vitro membranes, since in vivo fish membranes remain immersed in the natural environment.

-L464-477: Have authors some hypothesis on the differential sialylation and fucosylation of the analyzed mucins compared to fish gill mucins? Do they have any suggestion in order to modulate the glycosylation pattern so that their mucin model may better replicate the in vivo pattern?

In the same line, PAS+ goblet cells with neutral mucins are usually present in fish mucosal epithelia. Authors could also discuss the absence of those in their in vitro model.

-Authors do not consider the differential involvement of secreted and membrane-bound mucins in their study. Especially when mucins from cell lysates are analyzed, such different mucin types in addition to intracellular immature mucins associated to the Golgi apparatus, should be considered.

Besides, some minor corrections should be implemented:

- L49: Please, change for “fish mucosal epithelia are”.

- L108: Correct “RTgutGC”.

- L166: Please, correct “The medium … was”.

- L169: Correct “mucosal surfaces” to lower case.

- L204-234: Use µL for microliters.

- L218: Describe what “conc HAc” is.

- L225: Shift to “control media”.

- L251: Correct “MS/MS”.

- L310: Please, place “…, in appearance, …” between comas.

- Table1: Empty cells should be filled with 0 or dash.

- L393: Please, rewrite. E.g. “…their relative abundance was estimated…”

- L411: Please, rewrite. E.g. “… is incorporated similarly in both cell lines and Alcian blue staining was also similar in both”

- L412: Please, change “which were all”.

- L419: Correct “models that mimic”.

- L420: Correct “often result in”

- L425: Please, delete “of mucus”.

- L500: Correct “models present”.

Reviewer #2: The manuscript title “Mucin Producing In vitro Fish Mucosal Surfaces as a Model for Studying Host” by Quintana-Hayashi and colleagues is about the possible use of fish cell lines as in vitro mucosal surface models. These models pretend to be an alternative to in vivo experiments and be a model to investigate the effect of pathogens and modulatory components on mucin production in fish.

I found this article really interesting and a good alternative to reduce the number of fish in in vivo experiments. However, I found the manuscript difficult to read because the material and methods are not well explained and therefore the workflow is confusing. In addition, I missed a proper control in many experiments and statistical analysis. For all of these reasons, I consider that is needed deep changes in order to be published.

1-I do not know the protocol that authors used to differentiate fish cell lines into mucus producing cells. In material and methods it seems that only with culturing the cells in snapwell cell culture for 2 weeks and after that with the inhibitor DAPT and IWP-2 is enough but in the results is suggested that authors have used other methods to differentiate cell for example: line 278-282 “We started by investigating morphology, adherence, and mucin production in the salmonid cell lines RTgill-W1, CHSE-214 and SSE-5 under standard conditions as well as under conditions that we have used in the successful development of mammalian in vitro mucin producing mucosal surfaces from epithelial cells, including treatment of the cells with goblet cell differentiating media and prolonged culture ”. Please clarify these issue.

2-The workflow of the experiments is confusing, for example in the results of the sections: The metabolic label GalNAz is incorporated into both RTgill-W1 and CHSE-214 based in vitro membranes, O-glycan analyses from CHSE-214 and RTgill-W1 and A. salmonicida binds to RTgill-W1 and CHSE-214 cells. I do not know the protocol that authors used to differentiate fish cell lines if it is the same that in the section RTgill-W1 and CHSE-214 cell lines form an adherent layer and produce mucins. Please clarify this issue. I suggest to draw a schema in order to clarify how the experiments are performed.

3-Material and methods are incomplete; I do not know how the experiments are made and I missed commercial references of many reactives in many sections. In addition, I did not find the microscope use for the images taken in figure 1 and Figure2. In figure 2 C and F, I do not know how authors obtain these data and there are not properly explain in the legend. Moreover, in many sections of material and method, authors said for example aliquots were dot plot, cell lysate aliquots or paraffin embedded sections… but they do not indicate which samples were analyzed and therefore makes more difficult to understand the experiments that they perform.

4- In addition, I missed a proper control of many experiments. For example, in figure 2 I do not understand why the not labeled membranes are stained. I must include a positive and negative control to compare the results obtained and include also in C and F graphs. In figure 3 andfFigure 4 I would include a control of CHSE and RTGill without differentiate as a proper control in addition to the media control.

5- In figure 1 and Figure2 authors should include in both legends the magnification of these pictures and the number in the scale. In figure 2 please include statistical analysis of the graphs.

6-The binding assay in my opinion is not enough to confirm this issue. authors should include more experiments to confirm the binding.

7-The results showed in table 1, in my opinion, the spectrum of the peaks of the chromatogram of O-glycans should be included in the manuscript. I do not understand why some samples are dot blot and others by gel band, please explain properly.

8- I found some confusing sentences that I detailed below:

-line 80-82” A decreased secretion could lead to an increased amount of mucus in the cells, conversely, an increased secretion could lead to a decreased amount of mucus in the cells”. please clarify.

-line 461-463” the lower inter-individual variation and more distinct GalNAz staining of

the RTgill-W1 in vitro membranes most likely makes 462 this a better model for studies comparing effects of different conditions/treatment on mucin production”. I do not understand the sentence “more distinct GalNAz staining”, in the graph seems that there are not much variation in the fluorescence intensity. Please clarify this sentence.

-line 474-476”. Although the majority of the rainbow trout mucins are sialylated in line with the results from the RTGill-W1 mucin glycans, approximately 30% of the glycans are sialylated (6), a feature we did not detect among the RTGill-W1 mucin glycans”. It is contradictory the information in the sentence, first authors observed sialylated mucins but at the end of the sentence said they did not detect. Please clarify. In addition, line 487-488 “The high abundance of sialylated glycans is likely the reason that Aeromonas salmonicida bound to both cell lines”. Please clarify.

Minnor things

Line 33, DAPT define acronym.

Line 111-112 cldn3 and il-1β, il-6, il-8 and tnfα define acronym

Line 154 WNT acronym

Line 218, conc HAc define acronym

Line 298 figure 1, I will specify to Fig 1A, F in order to reference the information which is included in the sentence.

Line 307 AB define acronym

line 367 the bottom, I believe that it is on the top. Please change it.

-Indicate at least for the first time define h as hours

-Line 426 HT29-MTX-E12 define acronym and the specie of the cell line

-Line 508 Neu Ac define acronym

Reviewer #3: *General comments: The current manuscript proposes the use of fish cell lines as a potential model platform for evaluating host/pathogen interactions on mucosal surfaces by mucin production.

I think that the strengths of the study are the methods to differentiate cells and induce mucin production, which are quite clear and described in detail. Furthermore, considering the historical lack of tools and platforms at the phenotypic level in fish, this study also contributes to the in vitro characterization of the immune response in epithelial barriers. However, regarding its weaknesses, I think that a more quantitative or semi-quantitative evaluation of the different parameters evaluated was lacking, especially related to the production of mucins (Figure 1) and the capacity of the cell lysates to bind A. salmonicida (Figure 4). Finally, before proposing final acceptance of the manuscript there are aspects to improve, which I detail in the specific comments:

*Specific comments:

Title: The first part of the title could change from "Mucin Producing In vitro Fish Mucosal Surfaces" to "In vitro Fish Mucosal Surfaces Producing Mucins". This would improve the understanding of the title.

Keywords: "cell line" should be added

Abstract:

-Line 23: After "prevent disease and reduce antibiotic use." There is a sentence missing that connects this idea with the "Fish epithelia secrete mucus". For instance "To deal with this problem, fish epithelia that secrete mucus, which is mainly composed of highly glycosylated mucins, are a potential target for evaluation."

-Line 25: After "pathogens encounter." add "Therefore," the aim.. .

-Line 41: Change "These models thus" to "Thus, these models"

Materials and Methods:

-Line 138: Add "," after "flask"

-Line 175: In "H2O", the 2 must be in subscript.

-SSE-5 should be eliminated from the manuscript since due to their problems in adhering, the authors did not carry out the tests with these cells nor did they obtain data.

-Line 259: Performing blotting to determine the binding capacity of cell lysates to A. salmonicida does not seem to be the most sensitive test or the one that allows the best analysis. Considering the capacities that the authors have described, the test should have been carried out in 96-well plates with adherent capacity to determine OD, this would have allowed a quantitative experiment with better comparisons between the samples.

Results

-Line 277: RTgill-W1 and CHSE-214 cell lines form an adherent layer and produce mucins. The authors, in addition to showing the images (and describing them), should perform a semi-quantitative analysis (in software such as ImageJ) of the intensity and coverage of the labeling between the different samples. This will allow comparisons between cell lines, inducers and mucin production to be better established.

-From 278 to 286: This paragraph is just methodology again, it should be moved to that section (modifying it).

-From 327 to 331: Just methodology again, it should be moved to that section (modifying it)

-Seccion "The metabolic label GalNAz is incorporated into both RTgill-W1 and CHSE-214 based in vitro membranes". In figure 2, C and F should be merged and thus statistically evaluate the fluorescence intensity between RT-gill and CHSE214.

-From 348 to 350: Methodology again, it should be moved to that section (modifying it).

-Line 396: "A. salmonicida binds to RTgill-W1 and CHSE-214 cells" should be modified to the ability of the cell lysates to bind to the bacteria. For example: RTgill-W1 and CHSE-214 cell lysates binds A. salmonicida".

Discusssion:

Lines 485 and 487: Change "Aeromonas salmonicida" to "A. salmonicida"

Figures

In general, the resolution of the manuscript images that contain cells is low (even in the downloaded versions). This must be improved to achieve publication of the study.

References: Authors should consider reducing the percentage of self-citations (currently 60%). Of the 45 citations in the text, 27 are from one of the authors involved in this study. Multiple references in the same sentence should be eliminated and others from different groups incorporated, for example in the introduction.

6. PLOS authors have the option to publish the peer review history of their article (what does this mean?). If published, this will include your full peer review and any attached files.

Reviewer #1: No

Reviewer #2: No

Reviewer #3: **Yes: **Byron Morales-Lange

---

## [Author Response · Author response to Decision Letter 0]

30 Apr 2024

RESPONSE to REVIEWERS and EDITOR

We thank the editor and reviewers for providing many useful suggestions on how to improve our manuscript (new title: In vitro fish mucosal surfaces producing mucin as a model for studying host-pathogen interactions, PLOS ONE, PONE-D-23-19772). We have addressed all of the issues raised by the reviewers in our revised manuscript with a detailed description of each. We trust that the manuscript is now suitable for publication in PLoS ONE. Thank you for your continued consideration of this manuscript.

Sincerely,

Sara Lindén

Response to the EDITOR

Response: The manuscript has been formatted according to the guidelines, including title, author and affiliations, references, and manuscript body. The formatting changes are highlighted in yellow in the revised manuscript.

Response: We have removed the funding information from the manuscript.

3. We note that your Data Availability Statement is currently as follows: "All relevant data are within the manuscript and its Supporting Information files. Raw MS data on O-glycan analyses are available on glycopost ( https://glycopost.glycosmos.org/) using the project ID: GPST000334 (CHSE-214) and GPST000335 (RTgillW1). MSMS data with tentative structures are available https://unicarb-dr.glycosmos.org/references/523."

Response: All data are available. We have made a supplementary file to which we added the original photo of the gel from Figure 3 and the datapoints forming the basis for the graph in Figure 2. In addition, the MS data which can be accessed on https://glycopost.glycosmos.org/ using the project ID: GPST000334 (CHSE-214) and GPST000335 (RTgillW1) and MSMS data with tentative structures on available https://unicarb-dr.glycosmos.org/references/523." 

Response: The data is already uploaded on https://glycopost.glycosmos.org/ using the project ID: GPST000334 (CHSE-214) and GPST000335 (RTgillW1) and https://unicarb-dr.glycosmos.org/references/523. Once the manuscript is accepted it is a matter of minutes to make the data publicly accessible.

Response: we have removed the comments pertaining to the cell line that did not adhere as we did not take any photos of the empty plastic surface. 

We also removed the sentence “We tested labelling of mucins at 4 h post addition of GalNAz, however the label was not yet incorporated into the cells in sufficient amounts to be easily detected (data not shown).” 

Response: We have included the original uncropped photo of the gel shown in Figure 3 as supplementary Figure 1 and added (The uncropped image is available as Supplementary Figure 1) to the figure legend of Figure 3. 

Review Comments to the Author

Reviewer #1: This manuscript describes an interesting novel in vitro model based on modified fish cell lines for the study of mucin responses in mucosal epithelia. Its reading was inspiring and it will help to develop alternative methods to the in vivo ones.

There are some suggestions authors could follow in order to detail results and deepen discussion, which in my opinion could increase the study’s soundness.

-I wonder what the cell morphology in the in vitro membranes was, and if it changed after DAPT treatment. Could authors observe goblet cell morphology or cell polarization? Including a description on cell morphology, differentiation and polarization would be interesting. In fact, authors have included in the discussion section the issue of non-differentiation and non-polarization in unstimulated cultures on plastic surfaces, leaving this issue unanswered in the current model.

Response: We do not have data pertaining to this topic with the exception of the images provided. Therefore we removed the sentence mentioning polarization and apical from the discussion, although we did not claim that the current model had these. 

-Authors describe interindividual variation of GalNAz incorporation in terms of “relatively similar” or “larger variation” in each cell line. Statistical significance should be indicated in terms of an analysis of variance.

Response: We added statistics to the text: ‘There was no statistically significant difference in the level of GalNAz incorporation between the cell lines (p = 0.11). In the RTgill-W1 in vitro membranes, the level of incorporated GalNAz was relatively similar between membranes (Standard deviation, SD: 9.88, Figure 2C), while a tendency to larger variation between membranes was observed among the CHSE in vitro membranes (SD: 20.98, two sample F-test: p = 0.12, Figure 2F).’

Furthermore, we added a more detailed description in the figure legend, in which we also included the variance: ‘The quantified fluorescence intensity data sets for both cell lines passed the Kolmogorov-Smirnov test for normality. The sample variances (SD2) were 97.55 and 440.11 for RTgill-W1 and CHSE-214, respectively. The fluorescence intensity did not differ between RTgill-W1 and CHSE-214 (T-test, p = 0.11).’

-L430 and following: Authors could include a comment on the need of semi-wet interfaces for fish in vitro membranes, since in vivo fish membranes remain immersed in the natural environment.

Response: we have expanded the section so it now reads ‘Although fish are submerged in water most of the time we considered trying this method to ‘trick’ the cells to increase mucus production, however, the effects of a semi-wet interface with mechanical stimulation on mucin production could not be explored in the fish in vitro model here, as a rocking platform increased the incubator temperature by 12 °C, resulting in a suboptimal environment for the fish cells. Instead, the fish cells were cultured submerged in medium both on the apical and basolateral sides.’

-L464-477: Have authors some hypothesis on the differential sialylation and fucosylation of the analyzed mucins compared to fish gill mucins? Do they have any suggestion in order to modulate the glycosylation pattern so that their mucin model may better replicate the in vivo pattern?

In the same line, PAS+ goblet cells with neutral mucins are usually present in fish mucosal epithelia. Authors could also discuss the absence of those in their in vitro model.

Response: It is quite common that cell lines do not exactly mimic the glycosylation present in the epithelial site of origin, for example glycans can be truncated. We have added ‘If fucosylation is an important feature for planned experiments, it is possible to transfect cell lines with glycosyltransferases, including fucosyltransferases, to obtain a required glycoprofile, and included two references for this (https://www.nature.com/articles/s41467-021-24366-4

https://www.ncbi.nlm.nih.gov/pmc/articles/PMC1602278/) to the discussion section. In the part of the discussion pertaining to bacterial binding, we also added ‘That both cell lines carry epitopes A. salmonicida can bind to supports that they can be used to study host-pathogen interactions. For example, it could be interesting to investigate the role of specific glycans in host pathogen interaction by gene editing the cells for specific glycosyltransferases and compare the impact of presence/absence on pathogen binding and downstream host responses.’ We are not certain that PAS+ goblet cells are an abundant feature of gill epithelia in rainbow trout – In response to this comment we stained approximately 90 rainbow trout gills with PAS/AB, and only found rare PAS+ cells - we would say at least 95% of the cells were blue, very few were pink. We did not have any chinook salmon samples available, and did not find images of PAS+ by googling. Since we did not find clear evidence supporting that neutral goblet cells are a major feature of rainbow trout gills, we prefer to not write a discussion focusing on this. If the reviewer can point us to publications demonstrating that they are, we would be happy to incorporate them in the discussion, however, since we now have written about the possibility to gene edit the cell models to obtain different glycoprofiles and pointed to the lack of fucosylated glycans in the model, maybe this is sufficient?

-Authors do not consider the differential involvement of secreted and membrane-bound mucins in their study. Especially when mucins from cell lysates are analyzed, such different mucin types in addition to intracellular immature mucins associated to the Golgi apparatus, should be considered.

Response: We agree that what we analyzed was the glycan repertoire of the whole cell and not of individual mucins, or glycoforms of individual proteins. We do not deem it feasible to isolate different mucins separately for glycan analysis with the tools currently available for rainbow trout and chinook salmon. 

Besides, some minor corrections should be implemented:

- L49: Please, change for “fish mucosal epithelia are”.

Response: we have re-written as requested.

- L108: Correct “RTgutGC”.

Response: the missing ‘t’ has been added, thanks for detecting this error.

- L166: Please, correct “The medium … was”.

Response: we have corrected to medium.

- L169: Correct “mucosal surfaces” to lower case.

Response: we have corrected to lower case.

- L204-234: Use µL for microliters.

Response: we have replaced the u with µ.

- L218: Describe what “conc HAc” is.

Response: we have written out ‘acetic acid’.

- L225: Shift to “control media”.

Response: we have shifted to ‘control media’.

- L251: Correct “MS/MS”.

Response: we have corrected to MS/MS.

- L310: Please, place “…, in appearance, …” between comas.

Response: we have added the requested commas.

- Table1: Empty cells should be filled with 0 or dash.

Response: er have added the requested dashes.

- L393: Please, rewrite. E.g. “…their relative abundance was estimated…”

Response: we have re-written as requested.

- L411: Please, rewrite. E.g. “… is incorporated similarly in both cell lines and Alcian blue staining was also similar in both”

Response: the suggested sentence does convey the meaning that we tried to convey. Therefore we did not rewrite as suggested.

- L412: Please, change “which were all”.

Response: we have re-written as requested.

- L419: Correct “models that mimic”.

Response: we have re-written as requested.

- L420: Correct “often result in”

Response: we have re-written as requested.

- L425: Please, delete “of mucus”.

Response: we deleted as requested.

- L500: Correct “models present”.

Response: we removed the s as requested.

Reviewer #2: The manuscript title “Mucin Producing In vitro Fish Mucosal Surfaces as a Model for Studying Host” by Quintana-Hayashi and colleagues is about the possible use of fish cell lines as in vitro mucosal surface models. These models pretend to be an alternative to in vivo experiments and be a model to investigate the effect of pathogens and modulatory components on mucin production in fish.

I found this article really interesting and a good alternative to reduce the number of fish in in vivo experiments. However, I found the manuscript difficult to read because the material and methods are not well explained and therefore the workflow is confusing. In addition, I missed a proper control in many experiments and statistical analysis. For all of these reasons, I consider that is need

---

## [Decision Letter · Decision Letter 1]

29 Jul 2024

In vitro Fish Mucosal Surfaces  Producing  Mucin as a Model for Studying Host-Pathogen Interactions

PONE-D-23-19772R1

Dear Dr. Lindén,

We’re pleased to inform you that your manuscript has been judged scientifically suitable for publication and will be formally accepted for publication once it meets all outstanding technical requirements.

Kind regards,

Maria del Mar Ortega-Villaizan

Academic Editor

PLOS ONE

Additional Editor Comments (optional):

Reviewers' comments:

Reviewer's Responses to Questions

**Comments to the Author**

1. If the authors have adequately addressed your comments raised in a previous round of review and you feel that this manuscript is now acceptable for publication, you may indicate that here to bypass the “Comments to the Author” section, enter your conflict of interest statement in the “Confidential to Editor” section, and submit your "Accept" recommendation.

Reviewer #1: All comments have been addressed

Reviewer #2: All comments have been addressed

2. Is the manuscript technically sound, and do the data support the conclusions?

Reviewer #1: Yes

Reviewer #2: Yes

3. Has the statistical analysis been performed appropriately and rigorously? 

Reviewer #1: Yes

Reviewer #2: Yes

4. Have the authors made all data underlying the findings in their manuscript fully available?

Reviewer #1: Yes

Reviewer #2: Yes

5. Is the manuscript presented in an intelligible fashion and written in standard English?

Reviewer #1: Yes

Reviewer #2: Yes

6. Review Comments to the Author

Reviewer #1: Regarding the cell polarization matter, authors should feel free to include it in the discussion. I was curious about cell morphology and polarization observed during the experiment but did not pretend that it should be deleted. In fact, I encourage authors to deepen this amazing issue in future works.

In regard to the modulation of mucin glycosylation patterns of the in vitro mucosal models, authors address this topic reasonably by ssuggesting gene editing. In fact, since I do not work with salmonid species, my initial question on PAS+ neutral goblet cells in fish gills might have been out of context. I apologize.

In my opinion the M&M section and the provided figures include all necessary details and their quality and contrast are adequate.

Reviewer #2: The authors have incorporated the corrections I made to the manuscript and therefore I have been able to properly understand their work and consider it acceptable for publication.

7. PLOS authors have the option to publish the peer review history of their article (what does this mean?). If published, this will include your full peer review and any attached files.

Reviewer #1: No

Reviewer #2: No

---

## [Editor Report · Acceptance letter]

1 Aug 2024

PONE-D-23-19772R1 

PLOS ONE

Dear Dr. Lindén, 

I'm pleased to inform you that your manuscript has been deemed suitable for publication in PLOS ONE. Congratulations! Your manuscript is now being handed over to our production team.

Kind regards, 

on behalf of

Dr. Maria del Mar Ortega-Villaizan 

Academic Editor

PLOS ONE